# The Blood DNA Methylation Clock GrimAge Is a Robust Surrogate for Airway Epithelia Aging

**DOI:** 10.3390/biomedicines10123094

**Published:** 2022-12-01

**Authors:** Ana I. Hernandez Cordero, Chen Xi Yang, Xuan Li, Julia Yang, Tawimas Shaipanich, Julie L. MacIsaac, David T. S. Lin, Michael S. Kobor, Steve Horvath, Shu Fan Paul Man, Don D. Sin, Janice M. Leung

**Affiliations:** 1Centre for Heart Lung Innovation, St. Paul’s Hospital and University of British Columbia, Vancouver, BC V6Z 1Y6, Canada; 2Division of Respiratory Medicine, Department of Medicine, University of British Columbia, Vancouver, BC V5Z 1M9, Canada; 3Centre for Molecular Medicine and Therapeutics, University of British Columbia, Vancouver, BC V5Z 4H4, Canada; 4Department of Biostatistics, Fielding School of Public Health, University of California Los Angeles (UCLA), Los Angeles, CA 90095, USA; 5Department of Human Genetics, David Geffen School of Medicine, University of California Los Angeles (UCLA), Los Angeles, CA 90095, USA

**Keywords:** COPD, epigenetic age, airway epithelium, blood

## Abstract

One key feature of Chronic Obstructive Pulmonary Disease (COPD) is that its prevalence increases exponentially with age. DNA methylation clocks have become powerful biomarkers to detect accelerated aging in a variety of diseases and can help prognose outcomes in severe COPD. This study investigated which DNA methylation clock could best reflect airway epigenetic age when used in more accessible blood samples. Our analyses showed that out of six DNA methylation clocks investigated, DNAmGrimAge demonstrated the strongest correlation and the smallest difference between the airway epithelium and blood. Our findings suggests that blood DNAmGrimAge accurately reflects airway epigenetic age of individuals and that its elevation is highly associated with COPD.

## 1. Introduction

Chronic Obstructive Pulmonary Disease (COPD) is a common lung disease characterized by persistent airflow obstruction. One key feature of COPD is that its prevalence increases exponentially with age, even among non-smokers [1]. This has led some to speculate that COPD is a disease of accelerated aging. Because DNA methylation is a dynamic process with predictable changes that occur with aging, DNA methylation clocks have become powerful biomarkers to detect accelerated aging in a variety of diseases and can help prognose poor outcomes in severe COPD [2]. We have previously demonstrated that COPD is characterized by increased small airway epithelial DNA methylation age [3], but whether this is also reflected in peripheral blood is unclear. Identifying a more easily accessible blood biomarker of aging that can capture accelerated aging in the airway would provide a novel method of risk-stratifying patients with COPD at greatest risk for lung complications. Of the many currently available peripheral blood DNA methylation clocks, it is uncertain which ones perform best as a surrogate for airway aging. To explore our hypothesis that peripheral blood epigenetic age can accurately reflect airway epigenetic age, we investigated the linear relationship between blood and airway epithelial epigenetic age measurements of six epigenetic clocks.

## 2. Methods

Study cohort: We used paired small airway epithelial brushings and buffy coat blood samples from 44 participants in the St. Paul’s Hospital Bronchoscopy study cohort (University of British Columbia Research Ethics Board Certificates H11-02713 and H15-02166). Methods for bronchial epithelial cell collection have been previously described [3,4]. Cytologic brushings were obtained via flexible bronchoscopy in small airways with diameters < 2 mm from upper lobe segments. The study cohort consisted of 15 individuals with COPD and 29 individuals without COPD. COPD was diagnosed based on a physician diagnosis of COPD and a pre-bronchodilator forced expiratory volume in one second (FEV_1_)/forced vital capacity (FVC) ≤ lower limit than normal (LLN). We removed patients with bronchiectasis from downstream analyses (n = 2), and retained 42 patients (COPD = 15, non-COPD = 27).

DNA methylation profiling and epigenetic clocks: All samples were profiled for DNA methylation using the Illumina Infinium MethylationEPIC BeadChip microarray. The DNA methylation data were processed according to a previously described pipeline, which includes quality control and normalization steps [3,4]. We calculated the blood and airway epigenetic age based on six DNA methylation clocks by imputing DNA methylation profiles in the Horvath laboratory webtool (https://dnamage.genetics.ucla.edu/, accessed on 1 February 2022). Briefly, these clocks can be characterized as first and second generation. First generation clocks were developed based on DNA methylation markers that strongly correlate with chronological age and include DNAmAge (pan-tissue) [5], DNAmAgeHannum (blood-specific) [6], and DNAmAgeSkinBlood (fibroblast- and blood-specific) [7]. In addition to markers of chronological age, second generation clocks incorporated DNA methylation markers associated with various age-related phenotypes; for instance, DNAmPhenoAge [8], DNAmGrimAge [9], and DNAmTelomereLength (DNAmTL) [10] included DNA methylation markers that were highly correlated with clinical biomarkers of aging, inflammatory proteins and smoking, and telomere length, respectively. DNA methylation clocks are robust measurements of biological age as predictable methylation changes occur at specific regions along the genome during a lifespan, not only in response to age but also to environmental and disease insults [7]. Using the Horvath laboratory webtool we also estimated cell proportions in the blood samples according to methods by Houseman [11].

Correlation analyses: We first estimated the Pearson correlation between the chronological age and the DNA methylation age for each clock within each tissue. Second, we calculated the overall correlation between blood and airway DNA methylation age for each clock and explored these relationships within each disease group. Last, we constructed Bland–Altman plots to show the difference between the epigenetic clocks in blood and airway epithelium. Statistically significant correlations were defined at *p* < 0.05.

Airway and blood epigenetic age acceleration and COPD: Here, we defined epigenetic age acceleration as positive epigenetic age residuals obtained from the regression of an epigenetic clock on chronological age, which is interpreted as an older epigenetic age than expected based on chronological age. We tested the association between COPD and each epigenetic clock using the following linear model: epigenetic clock ~ COPD + age + sex + body mass index (BMI) + smoking status. Considering that DNA methylation can be affected by cell proportions, our blood analyses were adjusted for blood cell proportions (CD8 lymphocytes + CD4 lymphocytes + Natural killer cells + B cells + Monocytes + Granulocytes). Statistically significant results were defined at *p* < 0.05.

## 3. Results

Study cohort overview: The study cohort mean age and standard deviation was 67 ± 10 years and there were no significant differences in age between the COPD and non-COPD groups (*p* = 0.35). As expected, patients with COPD demonstrated lower lung function (FEV1% predicted = 66% [IQR = 55–79%]) compared to those without COPD (FEV1% predicted = 94% [IQR = 82–115%]) (*p* = 4.65 × 10^−4^). There was no significant difference in the number of individuals who smoke (*p* = 0.42) and sex (*p* = 0.16) between the two groups.

Blood and airway epigenetic age: Chronological age was highly correlated with DNA methylation age in blood. DNAmAgeSkinBlood showed the strongest correlation (R = 0.90, *p* = 1.06 × 10^−15^), followed by DNAmAgeHannum, DNAmAge, DNAmGrimAge, DNAmPhenoAge and DNAmTL (Table 1). For airway epithelial cells, of the six clocks, DNAmGrimAge demonstrated the strongest correlation with chronological age (R = 0.90, *p* = 3.86 × 10^−16^). The airway epithelial DNAmAge, DNAmAgeSkinBlood, DNAmAgeHannum, DNAmPhenoAge, and DNAmTL clocks demonstrated weaker correlations with chronological age compared to the blood estimates.

DNAmGrimAge showed the strongest correlation in epigenetic age between the blood and the airways (R = 0.93, *p* = 9.86 × 10^−19^), a relationship that was preserved across disease groups (Table 1). The between tissue correlation of DNAmGrimAge in the non-COPD group was R = 0.92 (*p* = 9.80 × 10^−12^), likewise the COPD group had a R = 0.91 *(p* = 2.87 × 10^−6^). Furthermore, DNAmGrimAge demonstrated the smallest difference between the blood and airway estimates. Figure 1 shows that the differences between blood and airway DNAmGrimAge are the closest to zero compared to the other five clocks. On the other hand, the between tissue correlation for DNAmAge, DNAmAgeHannum, DNAmAgeSkinBlood, DNAmPhenoAge, and DNAmTL varied substantially amongst the two groups, with DNAmTL demonstrating the weakest between tissue correlation in the COPD group (R = −0.005, *p* = 0.986).

COPD is associated with systemic and airway DNAmGrimAge:

We investigated the relationship between epigenetic age and COPD in both blood and airway samples. Out of the six epigenetic clocks, only DNAmGrimAge residuals were associated with COPD in both tissues (Figure 2 and Table 1). Other than DNAmGrimAge, no other clock had a significant association with COPD in blood, while in airway samples DNAmAgeHannum, DNAmAgeSkinBlood, DNAmPhenoAge, and DNAmTL residuals were also significantly associated with COPD. We further found that the blood and airway samples of COPD patients were epigenetically older compared to the patients’ chronological age (blood DNAmGrimAge [74 ± 6 years] vs. chronological age [70 ± 8 years] [*p* = 0.002], airway DNAmGrimAge [75 ± 6 years] vs. chronological age [70 ± 8 years] [*p*= 2.573 × 10^−50^]). On the other hand, blood samples of the non-COPD group show no significant differences between DNAmGrimAge and chronological age (*p* = 0.253), however their airways also demonstrated older epigenetic age (airway DNAmGrimAge [69 ± 8 years] vs. chronological age [66 ± 11 years]), albeit to a lesser degree (*p* = 0.002) than the COPD group.

## 4. Discussion

To our knowledge, this is the first report to examine the relationship between airway epithelial and blood epigenetic age in the same subjects. Our analysis suggests that of the six widely available DNA methylation clocks, DNAmGrimAge may be the most promising blood epigenetic age biomarker for assessing accelerated aging in the airways of individuals with and without COPD. DNAmGrimAge demonstrated strong consistency between blood and airway compartments. This is in agreement with our recent findings, where in two different cohorts of patients infected with human immunodeficiency virus, we showed that blood and airway DNAmGrimAge are associated with lung function [12]. In addition, we show here that the association of blood and airway age acceleration with COPD was only captured by the DNAmGrimAge clock. Because DNAmGrimAge was derived in part using methylation sites associated with smoking, we speculate that in comparison to the other DNA methylation clocks, DNAmGrimAge may have greater sensitivity in capturing methylation alterations in the airways. Conversely, our findings suggest that DNA methylation clocks that heavily rely on blood components (DNAmAgeHannum, DNAmPhenoAge, and DNAmTL) are weaker predictors of airway epithelial epigenetic age. Even DNAmAge, which was developed to accurately predict age in a wide range of cell types [5], and DNAmAgeSkinBlood, which is specific for blood and fibroblasts cells [7], showed only moderate correlations with chronological age in the airways. These clocks were mainly trained on blood samples and therefore may not be appropriate for assessing age acceleration in the airways of COPD patients. Limitations of our work include a small sample size. In addition, these results may not be generalizable to patients with other airways diseases such as asthma or bronchiectasis. It is possible that the cell composition of the airway samples could vary, however we have previously shown that our airway brushings contain a majority of epithelial cells [3], thus our findings mostly apply to airway epithelial cells. Finally, the correlation between whole lung tissue specimens and blood remains to be determined. Despite these limitations, our findings indicate that blood DNAmGrimAge accurately reflects the airway epigenetic age of individuals. Further work exploring the use of blood DNAmGrimAge as a biomarker for lung function decline and for prognostication in other pulmonary diseases is warranted.

## Figures and Tables

**Figure 1 biomedicines-10-03094-f001:**
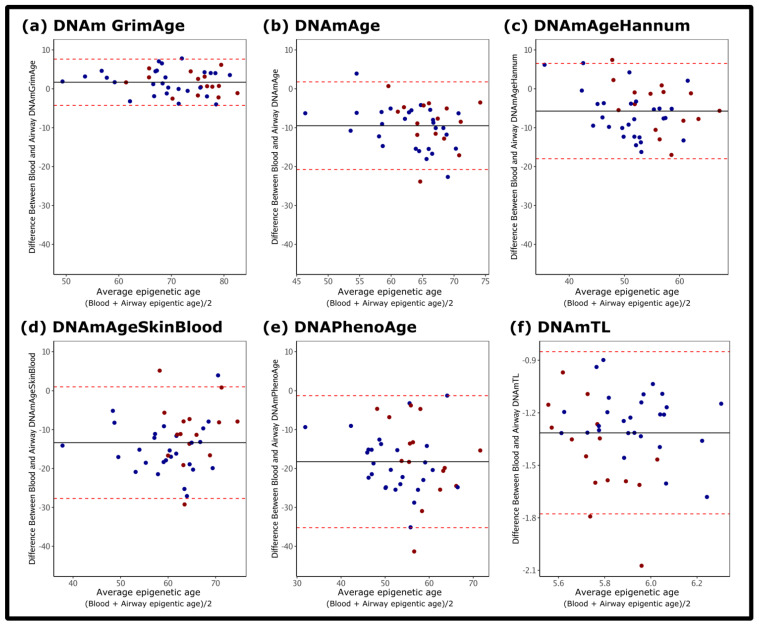
Difference between airway epithelial and blood epigenetic age. Bland–Altman Plots show the difference between the measurements (*y*-axis) and average values for the blood and airway epigenetic age (*x*-axis) for: (**a**) DNAmGrimAge; (**b**) DNAmAge (pan-tissue); (**c**) DNAmAgeHannum; (**d**) DNAmAgeSkinBlood; (**e**) DNAmPhenoAge; (**f**) DNAmTL. Blue dots: non-COPD. Red dots: COPD. Red dotted lines: standard deviation. Black horizontal line: average difference between blood and airway epigenetic age.

**Figure 2 biomedicines-10-03094-f002:**
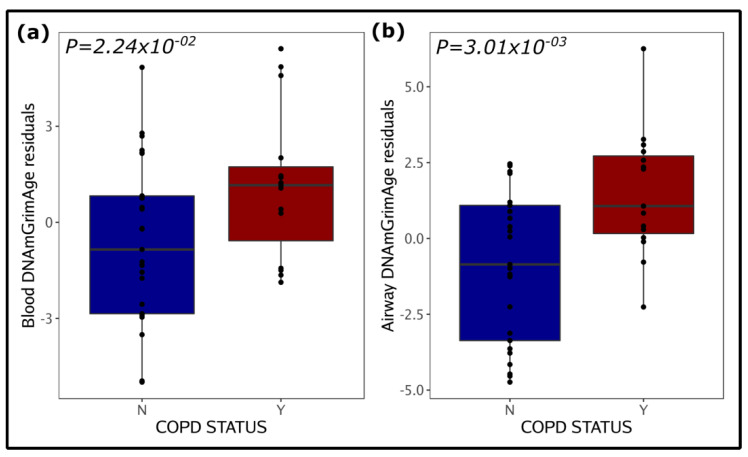
DNAmGrimAge age acceleration residuals in blood (**a**) and airway (**a**) samples and COPD. (**a**,**b**) show DNAmGrimAge age residuals based on COPD status (N = COPD negative, Y = COPD positive) (*x*-axis). The *y*-axis on the plots represents the residuals obtained from the regression of methylation age on the chronological age, body mass index, sex, and smoking status. Blood analysis was further adjusted for blood cell composition Significant *p*-values (*p* < 0.05) associated with the COPD status are shown inside the plots.

**Table 1 biomedicines-10-03094-t001:** DNA methylation age Pearson correlations between blood and airway epithelium.

DNA Methylation Clock	Group	R between Epigenetic Age and Chronological Age (*p*-Value)		R between Blood and Airway Epigenetic Age (*p*-Value)	COPD Test * (Estimate ^±^ and *p*-Value)	
		Blood	Airway		Blood	Airway
DNAmGrimAge	All	0.83 (1.62 × 10^−11^)	0.90 (3.86 × 10^−16^)	0.93 (9.86 × 10^−19^)	2.86 (2.24 × 10^−2^)	2.81 (3.01 × 10^−3^)
	COPD	0.85 (5.05 × 10^−5^)	0.93 (6.19 × 10^−7^)	0.91 (2.87 × 10^−6^)		
	Non-COPD	0.83 (1.04 × 10^08^)	0.91 (5.15 × 10^−11^)	0.92 (9.80 × 10^−12^)		
DNAmAge	All	0.86 (3.73 × 10^−13^)	0.71 (1.35 × 10^−7^)	0.59 (4.46 × 10^−05^)	1.05 (0.485)	2.37 (5.12 × 10^−2^)
	COPD	0.85 (5.06 × 10^−5^)	0.47 (7.42 × 10^−2^)	0.25 (0.360)		
	Non-COPD	0.85 (1.45 × 10^−8^)	0.79 (9.11 × 10^−7^)	0.67 (1.34 × 10^−4^)		
DNAmAgeHannum	All	0.87 (4.81 × 10^−14^)	0.55 (1.56 × 10^−4^)	0.63 (8.75 × 10^−6^)	1.46 (0.301)	4.66 (0.011)
	COPD	0.89 (1.06 × 10^−5^)	0.70 (3.86 × 10^−3^)	0.58 (2.29 × 10^−2^)		
	Non-COPD	0.88 (1.95 × 10^−9^)	0.50 (8.38 × 10^−3^)	0.61 (8.08 × 10^−4^)		
DNAmAgeSkinBlood	All	0.90 (1.06 × 10^−15^)	0.64 (4.09 × 10^−6^)	0.58 (5.89 × 10^−5^)	0.43 (0.767)	5.03 (2.92 × 10^−2^)
	COPD	0.82 (1.74 × 10^−4^)	0.47 (7.83 × 10^−2^)	0.13 (0.632)		
	Non-COPD	0.91 (2.43 × 10^−11^)	0.70 (4.82 × 10^−5^)	0.69 (7.92 × 10^−5^)		
DNAmPhenoAge ^+^	All	0.79 (5.74 × 10^−10^)	0.46 (1.98 × 10^−3^)	0.50 (7.80 × 10^−4^)	3.13 (0.107)	5.22 (3.13 × 10^−2^)
	COPD	0.78 (6.54 × 10^−4^)	0.44 (9.82 × 10^−2^)	0.15 (0.597)		
	Non-COPD	0.80 4.74 × 10^−7^)	0.44 (2.06 × 10^−2^)	0.58 (1.68 × 10^−3^)		
DNAmTL	All	−0.66 (1.62 × 10^−6^)	−0.23 (0.13)	0.40 (8.72 × 10^−3^)	0.05 (0.256)	−0.23 (4.44 × 10^−5^)
	COPD	−0.57 (2.78 × 10^−2^)	0.34 (0.200)	−0.005 (0.986)		
	Non-COPD	−0.72 (1.91 × 10^−5^)	−0.32 (0.110)	0.61 (6.62 × 10^−4^)		

***** Model: epigenetic clock ~ COPD + age, BMI, sex, smoking status; blood analyses were also adjusted for cell proportions, except for DNAmPhenoAge (^+^ COPD test was not adjusted for cell proportions because these are the main predictors embedded in the DNAmPhenoAge calculation). **^±^** Reference group: non-COPD; positive effects correspond to age acceleration.

## Data Availability

DNA methylation data will be deposited into GEO database upon manuscript publication. Metadata are available upon reasonable request directed to Janice Leung at Janice.leung.hli.ubc.ca.

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
