# Peer review of "The Blood DNA Methylation Clock GrimAge Is a Robust Surrogate for Airway Epithelia Aging"

_biomedicines, 2022, doi:10.3390/biomedicines10123094_

Round 1

Reviewer 1 Report

I have commentds below.

1,It is better to add one more figure.
2,Authors performed DNA methylation of COPD blood samples by ILLumina. They found age acceleration as epigenetic change. I do not understand they mentioned as "clock". They need to explain more detail.

Reviewer 2 Report

This study investigated which DNA methylation clock could best reflect airway epigenetic age when using blood samples. The findings suggests that blood DNAmGrim Age accurately reflects airway epigenetic age of individuals and that its elevation is highly associated with COPD.

The manuscript is well written and very clear for the most part. I only have some suggestions for Table1 and Figure 1.

Table 1: The 4th column title is confusing, based on the text the R values given are for epigenetic age (not epigenetic biomarkers?).

Figure1: I suggest putting the Clock Names above each sub-figure. Also, the X-axis title could be better. 
